# Functional morphology of the Cambrian archaeocyath sponge *Yukonensis*

**Zaid A. Qureshi**[1*◉], **Brandt M. Gibson**[2◉], **Simon A. F. Darroch**[3,4◉], **Marc Laflamme**[1◉]

**1** Chemical and Physical Sciences, University of Toronto Mississauga, Mississauga, Ontario, Canada, **2** Department of Agriculture, Geosciences, and Natural Resources, University of Tennessee at Martin, Martin, Tennessee, United States of America, **3** Senckenberg Institute and Museum of Natural History, Frankfurt am Main, Germany, **4** Cluster of Excellence (ESC 3121): TERRA - Terrestrial Geo-Biosphere Interactions in a Changing World, University of Tübingen, Tübingen

◉ These authors contributed equally to this work.
* zaid.qureshi@mail.utoronto.ca

## Abstract

Archaeocyatha is a diverse clade of Early to Middle Cambrian (Stage 2–4) reef building filter-feeding sponges. Their calcium carbonate skeletons allowed for a broad range of morphologies, from cylindrical to conical or even domal, representing a key adaptation for filter feeding across a wide range of environmental conditions. Within archaeocyath morphospace, *Yukonensis yukonensis* is uniquely constructed from multiple stacked, porous, subspherical chambers with prominent bowl-shaped thorny corollas surrounding the base of each chamber. Given its morphological complexity, *Yukonensis* has been challenging to interpret paleobiologically and integrate with our understanding of archaeocyath ecology as a whole. Here, we use fluid and structural mechanical modelling to provide insight on the biological utility of some of its unique anatomical characters. Our fluid dynamics modelling demonstrates that the thorny corolla substantially alters ambient flow preventing water from entering external tumuli pores that are present on the subspherical chambers. Our fluid-structure inter-action analyses show that the unique T-shaped ridges along the thorny corolla spines experience increased stress, implying that their architecture did not increase structural rigidity as previously hypothesized, but must instead have served an alternative function. Taken collectively, the results of our multiphysics modelling provide insight into the paleoecology of one of the most enigmatic Cambrian suspension feeders and a quantitative means of testing the efficacy of biological functions in the oldest animal reef communities.

## Introduction

Modern coral reefs are epicentres of diversity, supporting large numbers of animals, plants, and algae. While the lengthy geologic history of reefs dates back to the Archean [1,2], the evolution of calcifying skeletons in the latest Ediacaran and

**Data availability statement:** All 3D models (.x_b) and COMSOL Multiphysics (.mph) files are freely available at this Zenodo repository: https://doi.org/10.5281/zenodo.19458699.

**Funding:** BMG and ML were supported by the Dutch Research Council (NWO; grant number OCENW.M.21.031; https://www.nwo.nl/en/find-funding), and ML was supported by Natural Sciences and Engineering Research Council of Canada (2019-05405 ML; https://www.nserc-crsng.gc.ca/index_eng.asp). ZAQ was supported by the Natural Sciences and Engineering Research Council Undergraduate Student Research Award (https://www.nserc-crsng.gc.ca/students-etudiants/ug-pc/usra-brpc_eng.asp). Many of the techniques used in this research were developed thanks to joint funding from the US National Science Foundation (https://www.nsf.gov) and UK Natural Environment Research Council (NSF-NERC EAR2007928; https://www.ukri.org/councils/nerc/). The funders had no role in study design, data collection and analysis, decision to publish, or preparation of the manuscript. There was no additional external funding received for this study.

**Competing interests:** The authors report no competing interests.

early Cambrian reefs marked a shift to a more modern character that persists today [3–5]. During the early Cambrian (~528 Ma, "Tommotian", Terreneuvian, Middle–Late Cambrian Stage 2), Archaeocyatha – an extinct clade of calcifying sponges (Phylum Porifera, Class Archaeocyatha) [6–8] – became the first metazoan reef builders [9]. Alongside archaeocyaths, many Lower Cambrian reef frameworks also incorporated calcareous microbes [10], creating considerable variability in reef composition across environments and localities [11]. This structural complexity in turn supported biodiversity by creating niches for organisms in cryptic cavities and supporting complex trophic groups [11,12]. Due to their ability to facilitate reef-building, archaeocyaths were likely significant ecosystem engineers [13] and crucial components of early Cambrian ecosystems until their demise in the Cambrian Series 2, Stage 4 "Toyonian" (~510 Ma) [14–16]. Despite growing recognition of these roles during the main phase of the Cambrian Explosion [17], there remain outstanding questions about archaeocyath paleobiology and paleoecology.

Archaeocyaths exhibit a suite of distinctive anatomical features and growth forms that have proved difficult to interpret, particularly when comparisons are drawn with extant sponges [7,18,19]. In broad terms, archaeocyaths are characterized by relatively thick, porous, calcifying skeletons that range from cylindrical to conical and typically display a characteristic two-walled 'cone-in-cone' architecture [8]. The space between the walls is known as the intervallum and can contain a diverse array of calcium carbonate features that vertically and horizontally partition the skeleton of the archaeocyath [8]. Moreover, some archaeocyaths in the Order Monocyathida instead have a single wall and completely lack an intervallum [20]. Although these forms are often treated as the archetypal archaeocyath morphology, numerous taxa diverge markedly from this body plan [7,19–21].

*Yukonensis yukonensis* [22,23] (Fig 1) is an atypical archaeocyath known from the Cambrian Series 2, Stage 3 ("Botoman") from the Canadian Mackenzie Mountains [24], northern Canadian Rockies [25], eastern Alaska [21], and eastern Alabama [26]. Morphologically, it consists of a series of porous, double-walled and vertically stacked subspherical chambers (Table 1) [20,22]. The largest known specimen of *Y. yukonensis* includes at least eighteen chambers, while most specimens consist of five or fewer [21]. The outer wall is characterized by alternating rows of simple tumuli (Table 1) [21,22,24] that feed into the intervallum and subsequently the central chambers (Table 1) [21]. These chambers are interconnected to permit the internal flow of water across the chambers [21]. The most distinguishing feature of *Y. yukonensis* is the presence of repeated thorny corolla rings surrounding the junction between each chamber (Fig 1B-D). These external structures are composed of a series of curved, hollow spines (Fig 1D) that form the scaffold for thin, web-like calcium carbonate 'connecting membranes' superficially resembling an inverted umbrella or bowl [21]. Interestingly, a small subset of specimens known only from eastern Alabama have been reported to show secondary blunt protrusions perpendicular to the larger thorny corolla spines [26]. Spines and the connecting membrane are constructed from longitudinal T-shaped ridges (Fig 1D) that are thought to have either increased structural rigidity of individual spines [21], provided defence against predation [21], or altered the direction of water flow to aid in feeding [21].

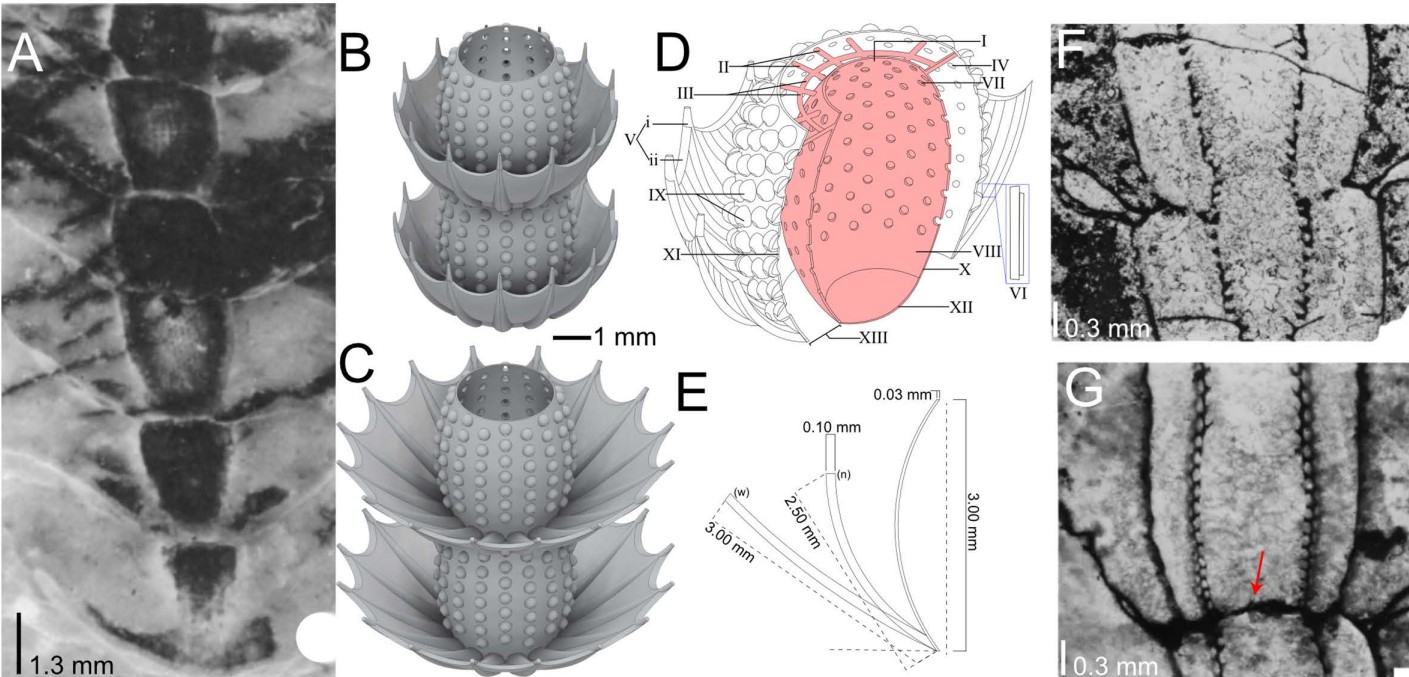

**Fig 1. Fossil specimens and reconstructions of *Yukonensis yukonensis* from the Mackenzie Mountains, Yukon, Canada.** A) Longitudinal section (from Handfield [22], Plate 23, Fig 4). B-C) Two chambers of a reconstructed 3D *Y. yukonensis* model with (B) narrow and (C) wide-angled thorny corolla. D) Cross-sectional diagram of a single chamber. The red colour indicates internal features that were not modelled (See Materials and Methods). The labels are as follows: I) Osculum, II) Rods, III) Synapticulae, IV) Outer wall pore, V) Thorny corolla, V (i) Spine, V (ii) Connecting membrane, VI) T-shaped ridge (enlarged), VII) Inner wall pore, VIII) Central cavity, IX) Tumuli, X) Inner wall, XI) Outer wall, XII) Vesicle, XIII) Intervallum. E) Two-dimensional view of the thorny corolla with labelled measurements of key features. The narrow thorny corolla is labelled with 'n', and the wide thorny corolla variant is labelled with 'w', F-G) Cross-section images from Handfield and McKinney [21] displaying the intervallum, the open central cavity between chambers (F; Plate 1, Fig 1), and the aporous horizontal vesicle membrane (red arrow – G; Plate 2, Fig 1). Sections A, F, and G reprinted from [21,22] under a CC BY license, with permission from Sedimentary Geology (SEPM), original copyright in 1967 and 1975, respectively.

**Table 1. Glossary of *Y. yukonensis* external morphology.**

| Terminology | Definition | Citation |
|---|---|---|
| Chamber | Each individual subspherical component that stacks to create the cup of *Yukonensis yukonensis*. | [20] |
| Osculum | Superior opening of the archaeocyath cup where water is exhaled and exits the organism. | [27] |
| Simple tumuli | Hollow mound-shaped pores on the outer wall with a lowered opening. | [19] |
| T-shaped ridge | Ornamentations surrounding the spines and connecting membrane of the thorny corolla that have a T-like shape. | [21] |
| Thorny corolla | A feature unique to *Y. yukonensis* that resembles an upside-down umbrella with spines and a connective carbonate membrane. It is connected to the outer wall of each chamber. | [20,22] |

Fluid dynamics is a technique that has been previously applied to archaeocyaths to gain a better understanding of their feeding mechanisms. Early biomechanical studies using flume and physical 3D models suggested that archaeocyaths were likely passive filter feeders [28,29], but these have been criticized for their use of inaccurate models and

experimental designs [30,31]. Instead, recent CFD studies on 3D-modelled single-walled monocyathid archaeocyaths [32] have supported an active filter feeding mechanism (i.e., flagellated pumping via an inhalant and exhalant canal system), given the demonstration in the CFD simulations of stagnant flow in the cavity, and drastically slower exhalent velocities when compared to modern sponges [33].

In this study, we examine how the distinctive external morphology of *Y. yukonensis* (tumuli and thorny corolla; Fig 1) [22] may have influenced its functional morphology and feeding efficacy compared to archetypical archaeocyathid sponges. Assessing the function of the enigmatic spines and corolla in *Y. yukonensis* is ideally suited to computational approaches. We use a novel combination of computational fluid dynamics (CFD) and fluid-structure interaction (FSI) analyses to reconstruct patterns of fluid flow around *Y. yukonensis* and use these data to constrain the function of key external anatomical features. CFD and FSI are well-suited for quantitatively assessing paleoecological hypotheses surrounding enigmatic and extinct organisms. While CFD is becoming increasingly common in paleontology, FSI analyses simulating the physical stresses applied upon a structure from a fluid are less so [34,35]. The combination of these two approaches in a paleontological dataset is therefore powerful as CFD permits the testing of hypotheses surrounding how organismal morphology interacted with fluid flow to aid biological and ecological function, while FSI examines to what extent these flow patterns would have exerted mechanical stress on different parts of the animal, and thus provide a means for assessing the feasibility of reconstructed life modes.

We address two previously proposed hypotheses surrounding the role of the thorny corolla and T-shaped ridges in the spines of *Y. yukonensis*: 1) that the thorny corolla was adapted to redirect flow to the external tumuli openings for feeding [21], and 2) that the T-shaped ridges evolved to increase structural rigidity, much as steel I-beams are used in construction [21]. These hypotheses generate testable predictions concerning fluid flow patterns and mechanical stress that are ideally suited to CFD/FSI. For example, if the thorny corolla aided feeding, fluid would be expected to be directed along the curvature of the corolla into the outer-wall tumuli. Conversely, a strong velocity gradient, fluid stagnation, and/or reduction of fluid being delivered to the tumuli within the thorny corolla could all indicate that the corolla impeded flow and therefore evolved for a different function. Likewise, if the T-shaped ridges (Fig 1D) enhanced structural rigidity in the spine and corolla, then stress build-up along the ridged spines would be predicted to be lower compared to spines lacking these ridges. By testing these hypotheses, we shed new light on the paleobiology of one of the most unusual archaeocyaths and ultimately help to understand the function of its strange features.

## Materials and methods

Digital models of *Y. yukonensis* were constructed and exported from Autodesk Fusion 360 and Autodesk Inventor Professional using anatomical measurements from collected photographs and published descriptions [20–22,24]. Detailed measurements can be found in Table 2. Due to the complex morphology and associated computational constraints, we targeted the specific anatomical regions at two scales: (i) a macroscopic specimen with four vertically stacked chambers with a focus on external morphology and (ii) a single spine with and without detailed longitudinal ridges. Note that the macroscopic model was constructed with four chambers that fall in the range of most preserved *Y. yukonensis* specimens [22]. This lower chamber count was necessary due to computational limitations, but it still permitted reconstructing flow across the chambers. Similarly, we neglected the blunt protrusions described in *Y. yukonensis* from eastern Alabama due to their limited distribution and potential taphonomic origin [26]. While ubiquitous model conditions are detailed here, the disparate scales of the morphological structures analyzed required individually targeted setups, which are subsequently described. No permits were required for the detailed study, which complied with all relevant regulations.

Interpretations of the paleoenvironmental conditions of *Y. yukonensis* vary among reported localities, reflecting the broad range of settings archaeocyaths could inhabit, from protected lagoons to deeper subtidal environments that are potentially below the photic zone [11]. For example, specimens from the Mackenzie Mountains in southeastern Yukon, Canada, likely lived in calm, low energy environments, inferred from the presence of peloidal micrite and the lack of

**Table 2. Comparisons between the *Y. yukonensis* dimensions reported in the literature and the dimensions of the reconstructed model.** Dimensions of the pores, tumuli, and spine distances vary slightly due to modelling and meshing constraints. Values marked by an asterisk only correspond to the wide-angled model.

| Morphological unit | Approximate *Y. yukonensis* dimensions based on literature | Dimensions of reconstructed model | Reference |
|---|---|---|---|
| Chamber diameter | 1.1 mm to 2.8 mm | 1.46 mm | [22] |
| Chamber length | 2.2 mm to 4.0 mm | 3.0 mm | [22] |
| Outer pore diameter | 0.15 mm to 0.20 mm | 0.10 mm | [22] |
| Tumuli diameter | 0.15 mm | 0.26 mm | [24] |
| Outer wall thickness | 0.03 mm to 0.04 mm | 0.03 mm | [24] |
| Tip-to-tip distance between opposing spines | 4.8 mm to 6.6 mm | Minimum distance: 4.6 mm/6.8 mm* | [22] |
| Number of spines | 13–16 | 13/16* | [22] |

fragmentation in most archaeocyath fossils [21,22,24]. In contrast, fossils found in the Jones Ridge Formation of eastern Alaska incorporate coarser spar, which suggests higher velocities at shallower depths [21,24,25]. Additionally, specimens from the Gataga River locality within the Rocky Mountains of northern British Columbia, Canada, have also been interpreted as occupying a shallow reef community [25]. To simulate these diverse marine environments, we replicated ambient flow conditions of 0.01, 0.03, 0.05, and 0.2 m s$^{-1}$, which fall within the range of modern shallow subtropical reefs [36,37].

All simulations were conducted using COMSOL Multiphysics 6.2. Simulations were solved for stationary solutions and meshes were determined by mesh refinement sensitivity analyses (S1 and S2 Tables). To determine the appropriate turbulence model, individual Reynolds numbers (*Re*) at a temperature of 293.15 K (20°C) were calculated for each setup using

$$Re = \frac{\rho_f U L}{\mu},$$

(1)

where $\rho_f$ is the density of the fluid (water = 998.20 kg m$^{-3}$), *U* is our maximum depth-averaged inlet velocity (0.2 m s$^{-1}$), *L* is the characteristic length scale of the height of the anatomical character of interest (*e.g.,* 0.012 m for the macroscopic model and 0.0025 m for the spine) or flow domain length (*e.g.,* 0.12 m for the macroscopic model and 0.00425 m for the spine), and $\mu$ is the dynamic viscosity (0.001093 kg s$^{-1}$ m$^{-1}$). All our individual Reynolds numbers fall within the laminar to transition zone; for consistency we used a laminar model for all simulations (S3 Table), which was necessary for the FSI simulations due to computational constraints.

## Spine models

We used FSI to assess fluid-derived stress distributions along the length of three hypothesized spine models. All spine models are approximately 2.5 mm in length depending on the angle. Spine A and Spine B are both curved cones with diameters (at the proximal attachment point) of 0.50 mm and 0.556 mm [21], respectively. Spine C includes T-ridges, with an outer ridge diameter of 0.556 mm and inner ridge diameter of 0.50 mm, corresponding to Spines B's and A's diameters. All spines are hollow in cross-section with a shell thickness of 0.05 mm. Spine B allowed for testing if differences in our results between Spine A and Spine C were due to the presence of the ridges and not merely the larger diameter in those regions. Since the angles at which the spines protruded out of living *Y. yukonensis* are unknown, three separate angles were chosen to investigate how this angle could affect stress based on fossil specimen photographs: 0º, 10º, and 15º to the body wall. See S1 Table for full details regarding the mesh sensitivity test of the spine.

Each spine model was placed in a hexahedral flow domain, with the spine attached to the downstream face. The hexahedral dimensions of this flow domain are 4 mm *x* 3 mm *x* 4.25 mm (W *x* D *x* H). Due to computational constraints, all spine models and flow domains were bisected in the middle along the Z-X plane, and a symmetry boundary condition was also applied on this plane (S1 Fig). The post-bisection dimensions for the flow domain are 4 mm *x* 1.5 mm *x* 4.25 mm (W *x* D *x* H). For the CFD setup, the spine and associated wall were prescribed no-slip conditions with a normal inflow inlet (e.g., constant inlet velocity along its surface) on the opposing face (S1 Fig). As stated earlier, a symmetry boundary condition was applied to the bisected wall of the flow domain. Open boundary conditions were prescribed on all remaining flow domain walls, and a pressure point constraint was selected as the lower right corner of the flow domain. The fluid physics coupling to the linear structural mechanics was one way, meaning that the deformation of the solid spine did not lead to the deformation of the flow field. A symmetry boundary condition was added on the bisected face of the spine for the structural mechanics component. A fixed constraint was added at the spine-wall interface. For numerical simplification, the spine's physical properties are assumed to be the same in all locations and were set to those reported for $CaCO_3$ [38]: density ($\rho_s$) = 2711 kg m$^{-3}$, Young's Modulus (*E*) = 88.197 GPa, and Poisson's Ratio (*v*) = 0.33.

## Macroscopic model

The macroscopic model of *Y. yukonensis* consists of four stacked porous chambers, each with a height of 3 mm cumulating to a total model height of 12 mm. A model with a total of four chambers was created based on most *Y. yukonensis* specimens preserving five or fewer chambers [22] and to reduce the number of meshing elements due to computational constraints. Similarly, only the outer wall of the organism was modelled to focus solely on the unique external structure of *Y. yukonensis*. The tumuli pores were geometrically simplified by modelling them as cone-shaped openings instead of hollow mounds to limit the presence of thin faces (as shown in Fig 1D cross-section). Two potential thorny corolla angles (Fig 1E) – one with a narrow thorny corolla (Fig 1B) and another with a wide morphology (Fig 1C) – were modelled to assess how corolla angle affected flow patterns. Mesh sensitivity tests were done on the narrow variant due to this version likely having greater interaction between the thorny corolla and outer wall. Models were placed in a hexahedral flow domain of dimensions 120 mm *x* 70 mm *x* 40 mm (W *x* D *x* H) (S2 Fig). Each model was placed in the center of the lower surface at one-third the streamwise distance from the inlet (40 mm) to prevent numerical anomalies arising from boundary conditions [39]. Similar to the FSI simulations, the flow domains and models were bisected along the Z-X center of the domain. The post-bisection flow domain dimensions were 120 mm *x* 35 mm *x* 40 mm (W *x* D *x* H). No slip conditions were applied to the *Y. yukonensis* model and the lower hexahedral face to which the model was attached (S2 Fig). Opposing downstream and upstream faces were prescribed outlet and fully developed inlet boundary conditions [34], respectively. All remaining boundaries parallel to the flow were prescribed as slip conditions. To reduce computational complexity, the far field domain received a coarser mesh than the near field domain and flow areas around the organism (S2 Fig). See S2 Table for mesh sensitivity test data. As with the spine FSI studies, a stationary laminar flow model was applied.

## Results

### Macroscopic CFD

Our CFD results for the macroscopic model of *Y. yukonensis* display several key characters of flow (labelled A through E in Fig 2). In all figures, flow field velocities have been normalized to the inlet velocity for each simulation, resulting in unitless velocities. The viscous sublayer, or laminar layer (Fig 2A), is characterized by a local low velocity region that approaches 0 near the no-slip lower surface. Downstream of the model, there is a decline in normalized fluid velocity to < 0.1 (Fig 2B). Following the downstream wake, the velocity begins to accelerate above 0.1 of the inlet. Directly upstream of the thorny corolla, there is an increase in fluid velocity that is most prominent on the uppermost thorny corolla (Fig 2C) and is reduced for the lowermost thorny corolla (Fig 2D). Following the exterior curvature of the thorny corolla, fluids are

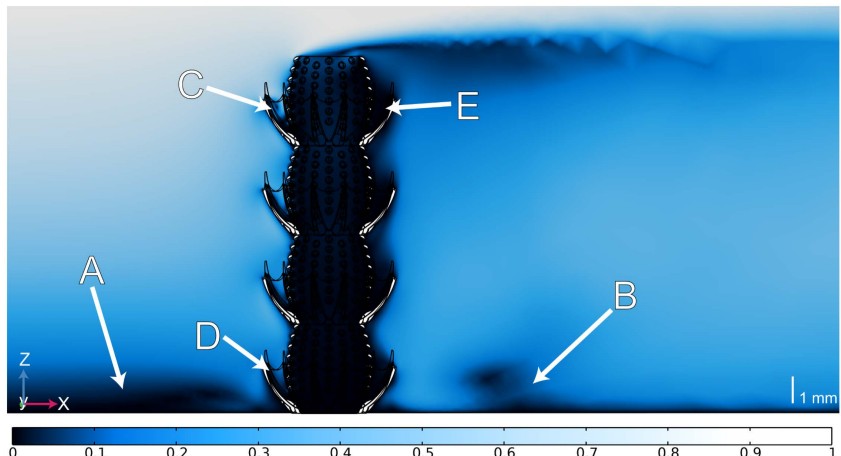

**Fig 2. CFD results for U = 0.2 m s⁻¹ inlet flow on the *Y. yukonensis* narrow macroscopic reconstruction.** Colour corresponds to the velocity normalized to the inlet. A) Viscous sublayer; B) Downstream wake; C) Acceleration of fluid entering upper thorny corolla; D) Reduced circulation within bottom-most thorny corolla; E) Downstream deceleration around thorny corolla.

partitioned vertically across chambers. In the space between the thorny corolla and the outer wall, flow is variable. For example, at the inner perimeter of the thorny corolla and outer wall, velocities are generally slow. However, at the middle of the thorny corolla (Fig 2C), there is an area of faster fluid flow that decelerates as it moves lower in the thorny corolla before interacting with the tumuli. In summary, the thorny corolla initially redirects water downwards, whereas flow above the corolla accelerates laterally to produce a pronounced vertical zonation (Fig 2).

Across all four tested velocities, there is little to no movement within the portion of the thorny corolla oriented downstream (Fig 2E; S3 and S4 Figs). This consistent deceleration is noticeable through all stacked chambers across *Y. yukonensis* – creating a 'dead zone' with normalized fluid velocities approaching ~0. For all velocities examined, this dead zone is most prominent in chambers found closer to the base of the organism (Fig 2; S3 and S4 Figs). Conversely, this dead zone becomes less pronounced in its vertical extent as the inlet velocity increases (S3 and S4 Figs).

These fluid circulation patterns were present in both the narrow (Fig 3A; S3 Fig) and wide (Fig 3B; S4 Fig) thorny corolla model simulations, however there are other differences in fluid flow velocities depending on the angle of the thorny corolla. In the upstream region of the wider corolla (Fig 3B), the increased clearance provides more room for water movement, promoting stronger circulation around the uppermost chambers. In contrast, the downstream region in the wider thorny corolla creates larger low velocities shadows (i.e., wake regions) along the outer perimeter of the thorny corolla compared to the narrow model (Fig 3). Despite these differences, corolla angle is demonstrated to have minimal effect on flow–tumuli interaction across chambers, since both the narrow and wide thorny corolla models show broadly comparable flow patterns (Fig 3).

## Spine FSI

Flow fields and associated von Mises (J2) stress distributions for all spine model variants (Fig 4; S5-S10 Figs) enable identification of regions most susceptible to failure or fracture. In total, three spine morphologies were modelled: Spine A is a smooth (non-ridged) spine, Spine B is a non-ridged spine with an enlarged diameter that accounts for the diameter change when ridges are present, and Spine C is the T-shaped ridge variant. For each model, the spines were tested at 0°, 10°, and 15° angles at inlet velocities of 0.01 m s⁻¹, 0.03 m s⁻¹, 0.05 m s⁻¹, and 0.2 m s⁻¹.

The minimum and maximum inflow velocities (U = 0.01 m s⁻¹ and 0.2 m s⁻¹) for the *Y. yukonensis* Spine C model (Fig 4) show that, although flow speeds differ, the spines consistently impede flow regardless of angle, causing water to

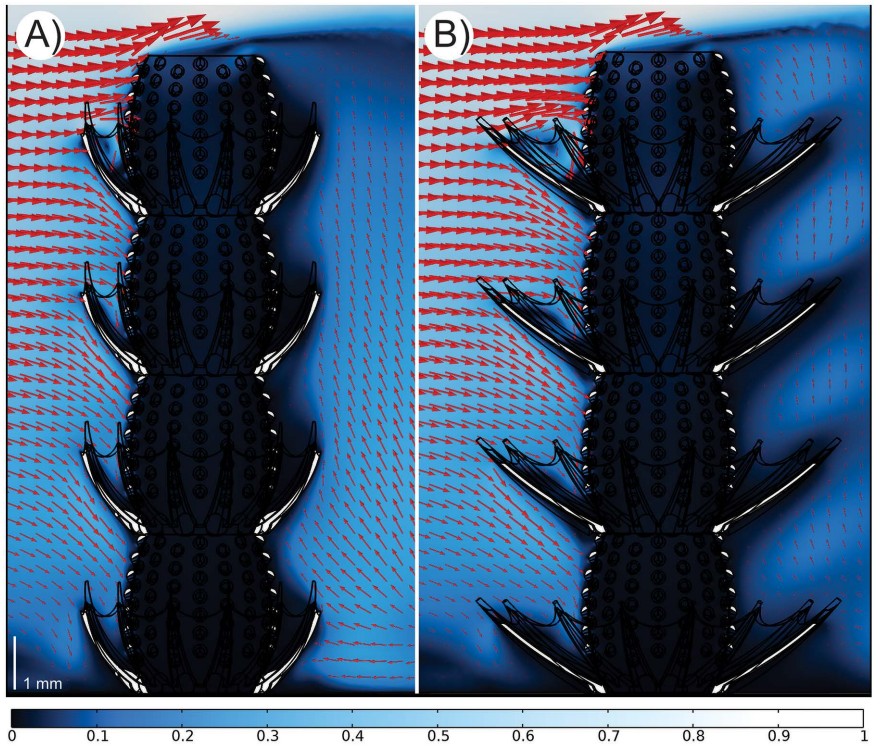

**Fig 3. Velocity profiles with volume arrows indicating the direction of fluid flow.** All profiles are at 0.2 m s$^{-1}$ with arrows scaled at 1.5x. A) Full view of the narrow model; B) Full view of the wide model under the same conditions. The velocity profile has been normalized with the inlet velocity.

decelerate upon contact. At 0° and 15° (normalized inlet velocity of 0.2 m s$^{-1}$), the slowest velocities consistently occur in the downstream wake region of the spine tips (Fig 4B, D) and approach ~0. At the minimum inlet velocity (0.01 m s$^{-1}$), the 0° model (and, to a lesser extent, the 15° model) exhibits reduced velocities extending from the spine tip towards the spine base (Fig 4). The moderate spine orientation (10°) shows slightly improved circulation between the outer wall and the spine (S7 Fig). These normalized velocity flow patterns are identical for all non-ridged spines across all velocities (S5 and S6 Figs). A viscous sublayer occurs along the distal no-slip surfaces of all spines.

The corresponding von Mises stress fields for the Spines A and B tests were plotted from their ventral, dorsal, and peripheral surfaces (S8-S10 Figs). The ventral section is the spine's downwards facing edge, the dorsal is the opposite upwards facing edge, and the peripheral is the centre of the spine surface. Our results show a steady increase in stress along the peripheral section from the spine tip to base; however, beyond approximately 2.25 mm from the spine tip, stress decreases before rising sharply (S8 and S9 Figs). Across all velocities, the 0° angle spine experiences less stress than the steeper angles and the disparity in stress between angles remain minimal (S8 and S9 Figs). Generally, the steepest angle (15°) experiences the greatest amount of stress. Moreover, at all angles investigated, higher velocities generally resulted in increased stress (S8 and S9 Figs). At a velocity of 0.2 m s$^{-1}$ and an angle of 15°, Spine A (S8 Fig) reaches its maximum stress of ~2600 N m$^{-2}$, while Spine B's (S9 Fig) maximum stress is ~2200 N m$^{-2}$. These maximum stress values are reached at the dorsal junction between the spine base and the outer wall of the organism. Ventral sections of Spines A and B also reach maximum stress at ~2.25 mm from the spine tip with values of ~1100 N m$^{-2}$ (S8 Fig) and ~650 N m$^{-2}$ (S9 Fig). For comparison, the peripheral section of Spines A and B under the same conditions reaches maximum stresses of ~340 N m$^{-2}$ and ~190 N m$^{-2}$ at ~2.25 mm from the tip, respectively. To summarize, the non-ridged spines (A and B) stress

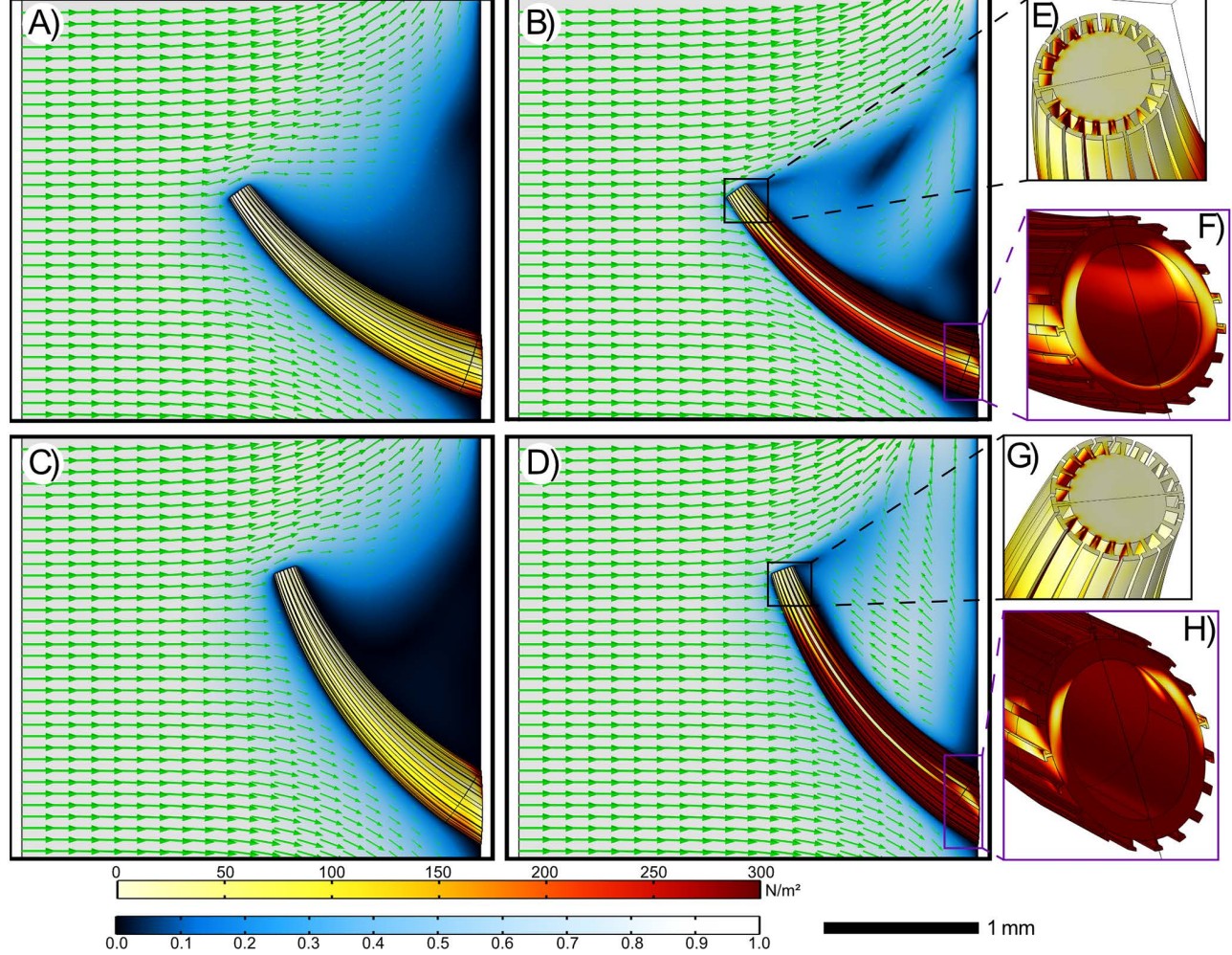

**Fig 4. Fluid-structure interaction analysis results for the ridged *Y. yukonensis* 0° and 15° Spine C models at 0.01 m s⁻¹ and 0.2 m s⁻¹ velocities that are normalized to the inlet.** All panels show the same flow field slice, and von Mises Stress distributions ($J_2$) are plotted along the spine surface. Green arrows represent the direction of the fluid's movement with length corresponding to normalized velocity: A) 0° angle spine at 0.01 m s⁻¹. B) 0° angle spine at 0.2 m s⁻¹. C) 15° angle spine at 0.01 m s⁻¹. D) 15° angle spine at 0.2 m s⁻¹. E) View of the spine's tip at 0° and 0.01 m s⁻¹ velocity. F) View of the spine's base at 0° and 0.2 m s⁻¹ velocity. G) View of the spine's tip at 15° and 0.01 m s⁻¹ velocity. H) View of the spine's base at 15° and 0.2 m s⁻¹ velocity. Scale corresponds to panels A, B, C, and D.

values are highest in the dorsal section and lowest in the peripheral section. As expected, stress scales with flow speed: the 0.2 m s⁻¹ case produces the highest stresses across all sections and angles, whereas the 0.01 m s⁻¹ case yields the lowest (S8-S10 Figs).

The ridged spine (Spine Model C; Fig 4; S7 and S10 Figs) also experiences stress values at the dorsal and ventral sections that are consistently greater when compared to Spines A and B. For example, at an inlet velocity of U = 0.2 m s⁻¹, peak values of approximately 3800 N m⁻² occur along the dorsal section where the wall-spine joint surrounding the T-shaped ridges is found (Fig 4F, H; S10 Fig). As with the non-ridged spines, the peripheral of Spine C experiences lower stress, while the highest stress is focused on the ridges at the spine base (Fig 4F, H). Across all angles, the highest inlet velocity generally produces the greatest peak stress in the dorsal and ventral sections (S10 Fig); however, stress in the peripheral section of Spine C can fall below that observed at the lowest velocities in the other spine models between ~0.5

mm to ~1.5 mm from the tip. Despite this, beyond ~1.5 mm, the stress begins to rise and quickly exceeds the maximum stress experienced by lower velocities. The maximum peripheral section stress at 15° reached by Spine C is ~260 N m$^{-2}$ (S10 Fig), while the maximum value for Spines A and B at the same angle and sections are ~340 N m$^{-2}$ and ~190 N m$^{-2}$ respectively (S8 and S9 Figs). The overall trends at Spine C's dorsal and ventral sections at all angles closely match those seen in Spines A and B but with greater stress values (S10 Fig).

## Discussion

The results of the CFD and FSI analyses provide an effective test for hypotheses surrounding the functional morphology of the unique thorny corolla and T-shaped ridges in *Y. yukonensis*. Specifically, we find that the morphology of the thorny corolla impeded fluid flow from entering the tumuli, thus making it an unlikely adaptation to facilitate passive suspension feeding. Likewise, we find that the morphology of T-shaped ridges neither re-direct flow towards tumuli, nor do they reduce mechanical stress in the spines under a range of realistic flow regimes. These observations allow us to propose new paleobiological reconstructions for *Y. yukonensis* that account for its unique morphology. Below, we first discuss the paleoecology of *Y. yukonensis* with a focus on the implications of the thorny corolla. Following this, we discuss the roles of the T-shaped ridges on the spines to propose plausible purposes for their presence.

### Paleoecology of *Yukonensis*

The presence of a thorny corolla in *Y. yukonensis*, which is unknown in any other archaeocyath species, implies a deviation from typical archaeocyath feeding behaviours. Based on our external reconstruction, the results (Figs 2 and 3) show tentative support for an active filter-feeding interpretation for *Y. yukonensis*. For passive feeding to be feasible, ambient flow must efficiently circulate into the central cavity via the tumuli and subsequently exit the organism, presumably through the osculum [29]. In contrast, active feeding involves the expenditure of energy such as the beating motion of flagellated units to pump water in and out of a sponge. In our simulations, we demonstrate that the presence of a thorny corolla accelerated flow along the thorny corolla's inner curvature, but advected flow away from the organism's tumuli (Fig 2C, D). Upon interaction with the thorny corolla, fluid is diverted and accelerated over the top of the structure, which then causes fluid to be redirected at high velocities downwards into the thorny corolla cup (Fig 3). These patterns were seen in both narrow and wide angled models, however, the wide model exhibited increased circulation across all velocities, likely caused by greater available space for fluid movement (S3 and S4 Figs). From these tests, there is little water transport from the thorny corolla into the tumuli; instead, most flow is channeled around the thorny corolla's interior, which at higher velocities creates a downstream vertical uplift of fluid out of the individual thorny corolla structures (Fig 3). However, we emphasize that these paleoecological interpretations are made from models lacking internal structures and anatomy, which may alter results.

Although computational constraints preclude detailed analysis of flow within the internal chambers, our results provide a basis for informed assumptions. In fossil specimens, the intervallum of *Yukonensis* includes a localized lattice of cylindrical rods cross-linked by synapticulae (i.e., short connecting bars) that occurs only at the uppermost portion of each chamber (Fig 1D) [24]. Because this framework does not extend continuously through the intervallum (Fig 1D), it likely had little influence on internal flow. The internal calcite rods and synapticulae may have provided structural support by bracing the intervallum and mechanically linking the inner and outer walls. Furthermore, they may have served as attachment sites for unfossilised soft tissues (i.e., choanocytes) that have yet to be reported in any archaeocyath fossil [19], and preclude their incorporation into the models.

Our external flow simulations show that the stacked nature of *Y. yukonensis* resulted in variable flow across each chamber (Figs 2 and 3). Handfield and McKinney [21] suggested that the lowermost chambers could have been metabolically inactive, with living tissue being restricted to the uppermost (distal) chambers. As shown in Handfield and McKinney [21], although rare, the intervallum and/or central cavity between chambers may be isolated by a horizontal calcium

carbonate membrane vesicle (formerly called 'dissepiment' [8]) (Fig 1F, G), which would have inhibited internal flow between chambers [24]. Because horizontal vesicles are not ubiquitous, they may have represented pathological growths formed in response to parasitism [24,40], structural damage [24,40], or they may have functioned as a means of restricting living tissues (and associated flow) to the uppermost chambers [21]. Our simulations support this interpretation, showing strong vertical flow stratification such that chambers nearest the substrate experience near-zero external flow (normalized velocities of ~0 in both wide and narrow thorny corolla variants) likely due to boundary-layer effects (Figs 2A and 3). Notably, at an inlet velocity of 0.2 m s$^{-1}$, the thorny corolla creates zones of downwards accelerations and upwards deceleration (Fig 3) that increase with the height of the chamber. At slower velocities (S3 and S4 Figs), this flow pattern is weaker but remains. Ultimately, these external flow results show a hydrodynamic pattern that would be improved by passive pressure gradients, active pumping, or both.

The functional role of vesicles remains unknown due to their inconsistent presence and geographic restrictions [24], possibly alluding to an environmental control. We propose that aporous vesicles could have provided biological advantages related to feeding and fluid movement in several ways. These aporous structures could have inhibited water exchange between chambers, therefore limiting internal fluid flow to each chamber with all excurrent flow being directed outward through the external pores and not via a central osculum [19,21]. However, this would be detrimental to internal flow by blocking the entry of water. As such, it is possible that vesicles separated the upper metabolically active regions found higher in the water column from those found closer to the substrate. Handfield and McKinney [21] also report cases where vesicles in *Yukonensis* block only the intervallum allowing the central cavity to remain open. Vesicles in the intervallum could serve as horizontal partitions that improve efficiency in fluid movement by guiding water out of the central cavity and reducing vertical flow up the intervallum; this impact could be similar to the aporous septa (vertical partitions in the intervallum) model tested by Savarese [29]. The Savarese model demonstrated that aporous vertical partitions in the intervallum become increasingly beneficial as an archaeocyath increases in height and faces faster current velocities by improving filtering efficiency [29]. For *Yukonensis,* we would anticipate that vesicles should become increasingly common in upper chambers for better filtering efficiency and reducing leakage out of pores. However, across *Y. yukonensis* specimens, the abundance and locations of vesicles appear to be random across chambers, which is also observed in other archaeocyaths [21,24]. Consequently, the growth of these vesicles has also been attributed to intervallum or central cavity damage (either physical or by parasitism; [21,40]). Based on our simulations, the stacked body plan of *Y. yukonensis* benefits the upper-most chambers, and water pumping may be improved in combination with vesicles assuming that they are efficiently placed.

**Functional role of ridged spines**

Exceptionally preserved *Y. yukonensis* specimens have thorny corolla spines striated by T-shaped ridges along their length [21]. Previous hypotheses [21,41] have suggested that the T-shaped ridges and spines could redirect fluid flow, provide structural reinforcement, be used as a defence against predation, improve stability in sediment, or increase surface area to be used by photosynthetic symbionts. Although some of these hypotheses are difficult to test, our FSI simulations can help investigate if they redirected flow or allowed for greater structural rigidity. The spines reduce fluid velocity downstream of the spine's distal tip, but the presence of T-shaped ridges appears to have little impact on redirecting fluid flow (Fig 4). For the ridges to aid in feeding, we would expect a noticeable redirection of flow between the ridges, increasing the velocity towards the ends of the spine closest to the wall attachment point (Fig 4; S10 Fig), and improving fluid delivery to the tumuli. Instead, we see that velocity patterns across the ridged and non-ridged spine models remain largely similar (Fig 4; S5-S7 Figs). The lack of velocity differences across ridged and non-ridged models suggest that the ridges have little impact on feeding.

Similarly, if the ridges were more structurally stable than the null model of a flat conical surface, we would expect lower overall stress values within and around the spine. Counterintuitively, our isolated spine simulations suggest that the

ridges would instead structurally weaken *Y. yukonensis* spines. The reduced stress at Spine C's peripheral transect is notable since it could be caused by increased drag from the presence of ridges (S10 Fig). However, this reduced stress only occurs in a small transect along the spine before stress increases closer to the spine base (S10 Fig). The dorsal and ventral weakening caused by ridges could lead to breakage along their lengths or at the junction with the outer wall where von Mises stresses were highest (Fig 4; S10 Fig). Thus, the presence of the ridges introduces a greater risk of potential damage likely disrupting the local flow. As such, it is unlikely that the purpose of the ridges would be either to improve flow or structural rigidity (assuming they lack non-fossilized soft tissue).

Hypothetically, the presence of soft tissue on the spines could have reduced the stress experienced by the calcium carbonate spine surface and served additional biological functions. The hollow structure of the spines suggests the presence of soft tissues that may have improved the structural integrity of the thorny corolla spines. For instance, the Cambrian sponge *Lenica sp.* has been described to have constructed spicules filled with organic material that were externally ornamented with grooves [42]. While archaeocyaths are aspiculate [8], the morphological similarities between the spines of *Yukonensis* and the spicules of *Lenica* are notable. Despite being independently derived, sea urchin spines also contain soft tissue that strengthens the calcite to make it more flexible despite being thin and hollow [43]. Alternatively, the T-ridges could create purposeful planes of weakness that break to prevent cracks forming throughout the entirety of the spine, thus protecting internal tissue, if present [43].

## Functional interpretation of the thorny corolla

Handfield and McKinney [21] suggested that the spines could have primarily functioned as additional attachments for flagellated units, allowing for actively guided fluid flow into the recesses of the thorny corolla and the tumuli. The ridges would have further increased the surface area for attachment along the spines, permitting additional flagellated units that could have been beneficial due to the organism potentially living at greater depths, with correspondingly slower ambient current velocities [21]. The external flagellated units, in combination with the velocity acceleration from the thorny corolla (Fig 3), could have improved internal circulation – especially when combined with presumed internal flagellated cells. This hypothesis requires the presence of the connecting calcium carbonate membrane portion of the thorny corolla (Fig 1D), though we acknowledge the connecting membrane is not consistently present in all fossil specimens [21]. If the hypothesis of Handfield and McKinney [21] is supported, external flagella along each spine could result in the movement of water without the connecting membrane portion of the thorny corolla. The beating of external flagella along the spine could also prevent the pooling of water within the thorny corolla by improving circulation (Fig 3). Metabolically active chambers of *Y. yukonensis* would have made use of the flagellated units, spines, and connecting membranes to increase overall flow. However, in the absence of fossilized soft tissues, these hypotheses remain difficult to test.

The thorny corolla could have helped support the vertically stacked morphology of *Yukonensis*. In areas with soft substrate, archaeocyaths have been shown to adopt a mushroom-shaped cup that widens as it grows [19,44], where these wider cups would have increased stability. Since *Y. yukonensis* is interpreted as primarily inhabiting deep-water settings characterized by soft substrates, this morphology may have been developed for soft substrate settings [24], with sediment accumulating around the lowermost chambers, similar to the "snowshoe effect" seen in soft substrate-specialized brachiopods to prevent sinking [21,45,46]. This function would have favoured wider-angled thorny corolla morphologies for increased stability. Moreover, the sediment would become trapped within the cup of the corolla, and maybe between the T-ridges as well, thus increasing overall stability. As a result, the thorny corolla may have been a necessary adaptation to facilitate the unique stacked morphology of *Y. yukonensis* on soft substrate.

## Conclusion

Our combination of CFD and FSI approaches provide a novel framework for systematically assessing the impact of unique anatomical characters in moving fluid environments and have provided key insights into the paleobiology of one

of the most morphologically enigmatic archaeocyath fossils yet discovered. Our multiphysics analyses have allowed us to systematically assess multiple hypotheses surrounding the functional role of unique morphology in *Y. yukonensis*. The advection of fluid around the tumuli and downstream within the thorny corolla indicates inadequate nutrient delivery inside the organism, supporting active feeding as seen in modern sponges. The FSI and CFD simulations suggest that the spines did not fundamentally alter fluid flow, and the T-shaped ridges do not improve structural reinforcement as previously hypothesized [21]. Instead, further studies should test to what extent the thorny corolla and ridges acted as attachments for potential external flagellated units that improved active filter feeding, or whether they increased stability in sediment through the snowshoe effect and ridges interlocking sediment.

## Supporting information

**S1 Table. Mesh sensitivity parameters for the spine of *Y. yukonensis*.** The mesh sensitivity test was conducted using Spine Model C at a 0º angle at an inlet velocity of 0.2 m s$^{-1}$. These conditions were chosen since they represent the most computationally intensive simulation among all three spine models. The selected mesh (Mesh 3) had 'Finer' near field flow domains and floor boundary, with the spine ridge boundary being 'Extra Fine'. The highlighted mesh was selected. These mesh parameters were also applied to the studies for Spine Models A and B.
(XLSX)

**S2 Table. Mesh sensitivity parameters for the macroscopic model of *Y. yukonensis*.** Analysis was conducted on the narrow macroscopic model at an inlet velocity of 0.2 m s$^{-1}$. For this test, four meshes were created, with Mesh 3 chosen for the studies. Mesh settings for all boundaries of the far field flow domain were set to be 'Normal', while the far field floor for Mesh 3 was prescribed the pre-defined 'Fine' settings. The near field domain boundaries, except for the floor, had a pre-defined resolution of 'Finer'. The surfaces of the model and near field floor were given a pre-defined resolution of 'Finer'. Each tumulus was selected and given a custom minimum element size of 0.02 mm. The thorny corolla was given a parameter of 'Extra Fine'. The pores of the model were given a custom minimum value of 0.08 mm with the remaining element size parameters being the pre-defined 'Finer' values. The highlighted mesh (Mesh 3) was selected.
(XLSX)

**S3 Table. Reynolds number calculations for all models.** Rho is the fluid density (kg m$^{-3}$), μ is the dynamic viscosity (kg s$^{-1}$ m$^{-1}$), and U is the depth-averaged inlet velocity (0.2 m s$^{-1}$). L is the length of the segment being measured in meters. Re is the final Reynolds Number and the regime is whether the system is laminar, transitional, or turbulent.
(XLSX)

**S1 Fig. Set-up diagram of spine FSI simulations.** Wall colour corresponds to the colour of the text. The symmetry wall is highlighted by a dark green outline. The pressure point constraint is visualized by the red dot on the symmetry wall's bottom right corner.
(TIF)

**S2 Fig. Set-up diagram of macroscopic CFD simulations.** The symmetry wall is highlighted by a dark green outline. The near field (NF) and wake domain (WD) are also labelled. The purple lines zoom in onto the macroscopic model of *Y. yukonensis*. The flow domain floor and model are no slip surfaces.
(TIF)

**S3 Fig. Normalized velocity profiles of the narrow model with volume arrows.** A) Velocity at 0.01 m s$^{-1}$; B) Velocity at 0.03 m s$^{-1}$; C) Velocity at 0.05 m s$^{-1}$; D) Velocity at 0.2 m s$^{-1}$. All arrows scaled by a factor of 1.5.
(TIF)

  

**S4 Fig. Normalized velocity profiles of the wide model with volume arrows.** A) Velocity at 0.01 m s$^{-1}$; B) Velocity at 0.03 m s$^{-1}$; C) Velocity at 0.05 m s$^{-1}$; D) Velocity at 0.2 m s$^{-1}$. All arrows scaled by a factor of 1.5.
(TIF)

**S5 Fig. Complete velocity profiles of Spine A.** Green arrows represent the direction of the fluid's movement with length corresponding to velocity magnitude (U). The von Mises Stress (N m$^{-2}$) is visualized on the spine, while the normalized velocity is visualized around the flow domain.
(TIF)

**S6 Fig. Complete velocity profiles of Spine B.** Green arrows represent the direction of the fluid's movement with length corresponding to velocity magnitude (U). The von Mises Stress (N m$^{-2}$) is visualized on the spine, while the normalized velocity is visualized around the flow domain.
(TIF)

**S7 Fig. Complete velocity profiles of Spine C.** Green arrows represent the direction of the fluid's movement with length corresponding to velocity magnitude (U). The von Mises Stress (N m$^{-2}$) is visualized on the spine, while the normalized velocity is visualized around the flow domain.
(TIF)

**S8 Fig. All graphs displaying the FSI results for Spine A.** Diagram on the right shows the locations where the stress profiles were measured. The von Mises Stress is measured in N m$^{-2}$. The coloured lines distinguish the four tested velocities at the three chosen angles and locations.
(TIF)

**S9 Fig. All graphs displaying the FSI results for Spine B.** Diagram on the right shows the locations where the stress profiles were measured. The von Mises Stress is measured in N m$^{-2}$. The coloured lines distinguish the four tested velocities at the three chosen angles and locations.
(TIF)

**S10 Fig. All graphs displaying the FSI results for Spine C.** Diagram on the right shows the locations where the stress profiles were measured. The von Mises Stress is measured in N m$^{-2}$. The coloured lines distinguish the four tested velocities at the three chosen angles and locations.
(TIF)

## Acknowledgments

The authors would like to thank the Department of Chemical and Physical Sciences at the University of Toronto Mississauga for allowing the use of their cluster to run the simulations. The authors also acknowledge the use of AI (Microsoft Copilot) to improve sentence structure and grammar.

## Author contributions

**Conceptualization:** Zaid A. Qureshi, Brandt M. Gibson, Simon A. F. Darroch, Marc Laflamme.

**Formal analysis:** Zaid A. Qureshi, Brandt M. Gibson.

**Funding acquisition:** Marc Laflamme.

**Investigation:** Zaid A. Qureshi.

**Methodology:** Zaid A. Qureshi, Brandt M. Gibson.

**Project administration:** Marc Laflamme.

**Resources:** Simon A. F. Darroch, Marc Laflamme.

**Supervision:** Brandt M. Gibson, Marc Laflamme.

**Validation:** Zaid A. Qureshi, Brandt M. Gibson.

**Visualization:** Zaid A. Qureshi.

**Writing – original draft:** Zaid A. Qureshi.

**Writing – review & editing:** Zaid A. Qureshi, Brandt M. Gibson, Simon A. F. Darroch, Marc Laflamme.

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
