## [Decision Letter · Decision Letter 0]

7 Jan 2026

PONE-D-25-63578Functional morphology of the Cambrian archaeocyath sponge YukonensisPLOS One

Dear Dr. Qureshi,

Thank you for submitting your manuscript to PLOS ONE. After careful consideration, we feel that it has merit but does not fully meet PLOS ONE’s publication criteria as it currently stands. Therefore, we invite you to submit a revised version of the manuscript that addresses the points raised during the review process.

We look forward to receiving your revised manuscript.

Kind regards,

Shamim Ahmad, PhD

Academic Editor

PLOS One

**Journal Requirements:**

1. When submitting your revision, we need you to address these additional requirements. Please ensure that your manuscript meets PLOS ONE's style requirements, including those for file naming. The PLOS ONE style templates can be found at https://journals.plos.org/plosone/s/file?id=wjVg/PLOSOne_formatting_sample_main_body.pdf and https://journals.plos.org/plosone/s/file?id=ba62/PLOSOne_formatting_sample_title_authors_affiliations.pdf 2. Please update your submission to use the PLOS LaTeX template. The template and more information on our requirements for LaTeX submissions can be found at http://journals.plos.org/plosone/s/latex. 3. In your manuscript, please provide additional information regarding the specimens used in your study. Ensure that you have reported human remain specimen numbers and complete repository information, including museum name and geographic location.  If permits were required, please ensure that you have provided details for all permits that were obtained, including the full name of the issuing authority, and add the following statement: 'All necessary permits were obtained for the described study, which complied with all relevant regulations.' If no permits were required, please include the following statement: 'No permits were required for the described study, which complied with all relevant regulations.' For more information on PLOS One's requirements for paleontology and archeology research, see https://journals.plos.org/plosone/s/submission-guidelines#loc-paleontology-and-archaeology-research. 4. We note that the grant information you provided in the ‘Funding Information’ and ‘Financial Disclosure’ sections do not match.  When you resubmit, please ensure that you provide the correct grant numbers for the awards you received for your study in the ‘Funding Information’ section. 5. Thank you for stating in your Funding Statement: BMG and ML were supported by the Dutch Research Council (NWO; grant number OCENW.M.21.031; https://www.nwo.nl/en/find-funding), and ML was supported by Natural Sciences and Engineering Research Council of Canada (2019-05405 ML; https://www.nserc-crsng.gc.ca/index_eng.asp). ZAQ was supported by the Natural Sciences and Engineering Research Council Undergraduate Student Research Award (https://www.nserc-crsng.gc.ca/students-etudiants/ug-pc/usra-brpc_eng.asp). Many of the techniques used in this research were developed thanks to joint funding from the US National Science Foundation (https://www.nsf.gov) and UK Natural Environment Research Council (NSF-NERC EAR-2007928; https://www.ukri.org/councils/nerc/). Sponsors had no role in influencing this study.  Please provide an amended statement that declares *all* the funding or sources of support (whether external or internal to your organization) received during this study, as detailed online in our guide for authors at http://journals.plos.org/plosone/s/submit-now. Please also include the statement “There was no additional external funding received for this study.” in your updated Funding Statement. Please include your amended Funding Statement within your cover letter. We will change the online submission form on your behalf. 6. Thank you for stating the following financial disclosure: BMG and ML were supported by the Dutch Research Council (NWO; grant number OCENW.M.21.031; https://www.nwo.nl/en/find-funding), and ML was supported by Natural Sciences and Engineering Research Council of Canada (2019-05405 ML; https://www.nserc-crsng.gc.ca/index_eng.asp). ZAQ was supported by the Natural Sciences and Engineering Research Council Undergraduate Student Research Award (https://www.nserc-crsng.gc.ca/students-etudiants/ug-pc/usra-brpc_eng.asp). Many of the techniques used in this research were developed thanks to joint funding from the US National Science Foundation (https://www.nsf.gov) and UK Natural Environment Research Council (NSF-NERC EAR-2007928; https://www.ukri.org/councils/nerc/). Sponsors had no role in influencing this study.   Please state what role the funders took in the study.  If the funders had no role, please state: "The funders had no role in study design, data collection and analysis, decision to publish, or preparation of the manuscript." If this statement is not correct you must amend it as needed. Please include this amended Role of Funder statement in your cover letter; we will change the online submission form on your behalf. 7. In the online submission form, you indicated that your data will be submitted to a repository upon acceptance.  We strongly recommend all authors deposit their data before acceptance, as the process can be lengthy and hold up publication timelines. Please note that, though access restrictions are acceptable now, your entire minimal  dataset will need to be made freely accessible if your manuscript is accepted for publication. This policy applies to all data except where public deposition would breach compliance with the protocol approved by your research ethics board. If you are unable to adhere to our open data policy, please kindly revise your statement to explain your reasoning and we will seek the editor's input on an exemption. 8. When completing the data availability statement of the submission form, you indicated that you will make your data available on acceptance. We strongly recommend all authors decide on a data sharing plan before acceptance, as the process can be lengthy and hold up publication timelines. Please note that, though access restrictions are acceptable now, your entire data will need to be made freely accessible if your manuscript is accepted for publication. This policy applies to all data except where public deposition would breach compliance with the protocol approved by your research ethics board. If you are unable to adhere to our open data policy, please kindly revise your statement to explain your reasoning and we will seek the editor's input on an exemption. Please be assured that, once you have provided your new statement, the assessment of your exemption will not hold up the peer review process. 9. We note that Figure 1 in your submission contain copyrighted images. All PLOS content is published under the Creative Commons Attribution License (CC BY 4.0), which means that the manuscript, images, and Supporting Information files will be freely available online, and any third party is permitted to access, download, copy, distribute, and use these materials in any way, even commercially, with proper attribution. For more information, see our copyright guidelines: http://journals.plos.org/plosone/s/licenses-and-copyright. We require you to either present written permission from the copyright holder to publish these figures specifically under the CC BY 4.0 license, or remove the figures from your submission: a. You may seek permission from the original copyright holder of Figure 1 to publish the content specifically under the CC BY 4.0 license.  We recommend that you contact the original copyright holder with the Content Permission Form (http://journals.plos.org/plosone/s/file?id=7c09/content-permission-form.pdf) and the following text:“I request permission for the open-access journal PLOS ONE to publish XXX under the Creative Commons Attribution License (CCAL) CC BY 4.0 (http://creativecommons.org/licenses/by/4.0/). Please be aware that this license allows unrestricted use and distribution, even commercially, by third parties. Please reply and provide explicit written permission to publish XXX under a CC BY license and complete the attached form.” Please upload the completed Content Permission Form or other proof of granted permissions as an "Other" file with your submission.  In the figure caption of the copyrighted figure, please include the following text: “Reprinted from [ref] under a CC BY license, with permission from [name of publisher], original copyright [original copyright year].” b. If you are unable to obtain permission from the original copyright holder to publish these figures under the CC BY 4.0 license or if the copyright holder’s requirements are incompatible with the CC BY 4.0 license, please either i) remove the figure or ii) supply a replacement figure that complies with the CC BY 4.0 license. Please check copyright information on all replacement figures and update the figure caption with source information. If applicable, please specify in the figure caption text when a figure is similar but not identical to the original image and is therefore for illustrative purposes only. 10. Please upload a new copy of Figures 4, 5, S8, S9, S10, S11, S12 and S13 as the detail is not clear. Please follow the link for more information:  https://journals.plos.org/plosone/s/figures 11. If the reviewer comments include a recommendation to cite specific previously published works, please review and evaluate these publications to determine whether they are relevant and should be cited. There is no requirement to cite these works unless the editor has indicated otherwise.

**Additional Editor Comments:**

Dear Author,

The reviewer finds the application of computational fluid dynamics (CFD) to Archaeocyathus yukonensis yukonensis to be innovative and potentially valuable. However, they raise substantial concerns that currently undermine the interpretability and robustness of the study, necessitating a major revision before further consideration.

The primary concern relates to the organization and clarity of the manuscript. At present, the biological morphology of Yukonensis is insufficiently described and visualized before the introduction of modeling parameters. This makes it difficult for readers particularly paleontologists to understand how the numerical geometry relates to the fossil organism. A clearer, more systematic morphological description, supported by figures that explicitly illustrate key structures (e.g., corolla morphology, spine arrangement, chamber architecture), should precede the CFD methodology.

More critically, the reviewer identifies fundamental discrepancies between the fossil morphology and the geometry used in the simulations, which may significantly affect the validity of the flow results:

• The absence of the inner wall in the CFD model is problematic given the sponge-grade organization of archaeocyaths, where flow dynamics are governed by pressure gradients across the intervallum. The rationale provided for excluding the inner wall is insufficient, and its omission likely alters internal flow resistance in a way that compromises interpretations of feeding strategy.

• The representation of the thorny corolla as two-dimensional, zero-thickness planes oversimplifies structures that are described and illustrated as thick, three-dimensional spines. Such simplification removes surface rugosity and turbulence effects that are likely critical for flow mixing and pore-scale dynamics.

• The simplification of curved, hollow spines into straight cones eliminates the very geometric features that could plausibly redirect flow, which is a central hypothesis of the study.

• The reduction of chamber number in the full-body model (four chambers versus at least six observed in specimens) raises concerns about boundary effects and whether the reported “dead zones” reflect biological reality or model artifacts. Justification or sensitivity testing is required.

Because of these geometric simplifications, the reviewer cautions that the conclusion—interpreting Yukonensis as an active feeder based on stagnant passive flow—may be model-dependent rather than biologically driven. The conclusions should therefore be significantly softened, clearly acknowledging that passive flow inefficiency applies only under the specific modeled constraints.

In addition, the reviewer notes specific technical issues that must be addressed:

• The reported Young’s modulus value for spines appears inconsistent with known values for calcite and may represent a typographical error.

• There are discrepancies between spine spacing measurements in Table 2 and those used in the model, which require verification against the fossil material.

Overall, while the study has promise, substantial revisions to manuscript structure, morphological fidelity, modeling justification, and interpretive claims are required. Addressing these points carefully will significantly strengthen the manuscript.

With warm Regards,

Dr S Ahmad

Reviewers' comments:

Reviewer's Responses to Questions

**Comments to the Author**

1. Is the manuscript technically sound, and do the data support the conclusions?

Reviewer #1: Partly

2. Has the statistical analysis been performed appropriately and rigorously? 

Reviewer #1: N/A

3. Have the authors made all data underlying the findings in their manuscript fully available?

Reviewer #1: Yes

4. Is the manuscript presented in an intelligible fashion and written in standard English?

Reviewer #1: Yes

5. Review Comments to the Author

**Reviewer #1:** This manuscript employs computational fluid dynamics modeling to test feeding strategies and structural stability in the enigmatic archaeocyath Yukonensis yukonensis. While the application of numerical modeling to this unique morphology is valuable, I have significant concerns regarding 1) the structure of the article, and 2) critical discrepancies between the actual fossil morphology and the geometry used for the simulation. I therefore suggest a major revision for the manuscript before publication. Of note, although I am familiar with archaeocyath, I am not a specialist in this specific taxon nor in functional morphology modeling, so I invite the authors to correct me if my interpretation of the model setup is mistaken.

1. It is difficult to follow the morphology of Yukonensis based on the current figures and text structure. Fig 1B, C does not clearly show the spine structure, yet the text describes model parameters before adequately visualizing the biological structure. As a reader, I found it confusing to reconcile the "wide" vs. "narrow" corolla descriptions (Fig 1A vs 1D) with the single species description. As a paleontologist, I believe it is necessary to fully describe the reconstructed biological structure of Yukonensis before introducing the modeling parameters, a step that is currently missing in the manuscript.

2. There are critical discrepancies between the organism described in the text/images and the 3D geometry used for the simulations, which likely compromise the validity of the flow model results.

- The characteristic double-walled structure of archaeocyaths is clearly visible in the photomicrographs (Fig 1E, F) and the diagram (Fig 1C). However, the model (Fig 1B) lacks an inner wall entirely. The authors state this was done to "focus solely on the external structure" (lines 183-184), but for a sponge-grade organism, flow is dictated by the pressure gradient across the intervallum (Savarese, 1992). Removing the inner wall would fundamentally alter the internal flow resistance. This is particularly important given that the authors are interpreting the organism as an active filter feeder.

- The authors model the complex thorny corolla as "two-dimensional planes with no thickness" (Lines 182-183). However, the text describes the corolla as an "inverted umbrella" with distinct spines, and the authors note spines are up to 0.5 mm in diameter (Line 158, Fig. 1E). Modeling such large structures with zero thickness treats the corolla as a smooth baffle. In reality, thick spines would create surface rugosity and turbulence, potentially mixing flow into the pores, a mechanism completely lost in the simplified model.

- While the text acknowledges "curved, hollow spines" (and Fig 1E shows a clavate/club-like shape), the methodology simplifies them into straight cones. Given that the study aims to test flow redirection, straightening a curved feature removes the very trait (curvature) that would most likely redirect fluid.

- The full-body model uses only four chambers, whereas Fig. 1A shows at least six. The flow results (Fig 2) show significant differences between the top and bottom chambers. The authors should justify why four chambers are sufficient to represent a mature organism or conduct a sensitivity test with more chambers to ensure the observed "dead zone" is not simply an inlet/outlet boundary artifact.

- The authors conclude Yukonensis was an active feeder because the passive model resulted in stagnant flow. However, as discussed above, because the model oversimplifies the morphology of the organism, this result may be an artifact of the geometry rather than the biology. The conclusion should be softened to state that passive flow is inefficient under these specific geometric constraints.

3. Minor Comments

- Line 179 lists the Young’s Modulus of the spine as 8897 GPa. The Young’s Modulus of calcite is typically ~70–80 GPa. Is this a typo for 88.97 GPa?

- Table 2 states the distance between spines is 4.8–6.6 mm, with 13 to 16 spines. This seems disproportionately large compared to the chamber size and differs significantly from the dimensions used in the model (1.105–1.358 mm). Please verify if these dimensions are correct relative to the fossil specimens.

6. PLOS authors have the option to publish the peer review history of their article (what does this mean?). If published, this will include your full peer review and any attached files.

Reviewer #1: **Yes:** Jeong-Hyun Lee

---

## [Author Response · Author response to Decision Letter 1]

30 Mar 2026

Journal Requirements:

3. In your manuscript, please provide additional information regarding the specimens used in your study. Ensure that you have reported human remain specimen numbers and complete repository information, including museum name and geographic location.

CHANGES MADE: Since no permits were required for this study, this statement has been added in Lines 146 – 147.

For more information on PLOS One's requirements for paleontology and archeology research, see https://journals.plos.org/plosone/s/submission-guidelines#loc-paleontology-and-archaeology-research.

5. Thank you for stating in your Funding Statement:

BMG and ML were supported by the Dutch Research Council (NWO; grant number OCENW.M.21.031; https://www.nwo.nl/en/find-funding), and ML was supported by Natural Sciences and Engineering Research Council of Canada (2019-05405 ML; https://www.nserc-crsng.gc.ca/index_eng.asp). ZAQ was supported by the Natural Sciences and Engineering Research Council Undergraduate Student Research Award (https://www.nserc-crsng.gc.ca/students-etudiants/ug-pc/usra-brpc_eng.asp). Many of the techniques used in this research were developed thanks to joint funding from the US National Science Foundation (https://www.nsf.gov) and UK Natural Environment Research Council (NSF-NERC EAR-2007928; https://www.ukri.org/councils/nerc/). Sponsors had no role in influencing this study.

6. Thank you for stating the following financial disclosure:

BMG and ML were supported by the Dutch Research Council (NWO; grant number OCENW.M.21.031; https://www.nwo.nl/en/find-funding), and ML was supported by Natural Sciences and Engineering Research Council of Canada (2019-05405 ML; https://www.nserc-crsng.gc.ca/index_eng.asp). ZAQ was supported by the Natural Sciences and Engineering Research Council Undergraduate Student Research Award (https://www.nserc-crsng.gc.ca/students-etudiants/ug-pc/usra-brpc_eng.asp). Many of the techniques used in this research were developed thanks to joint funding from the US National Science Foundation (https://www.nsf.gov) and UK Natural Environment Research Council (NSF-NERC EAR-2007928; https://www.ukri.org/councils/nerc/). Sponsors had no role in influencing this study.

7. In the online submission form, you indicated that your data will be submitted to a repository upon acceptance. We strongly recommend all authors deposit their data before acceptance, as the process can be lengthy and hold up publication timelines. Please note that, though access restrictions are acceptable now, your entire minimal dataset will need to be made freely accessible if your manuscript is accepted for publication. This policy applies to all data except where public deposition would breach compliance with the protocol approved by your research ethics board. If you are unable to adhere to our open data policy, please kindly revise your statement to explain your reasoning and we will seek the editor's input on an exemption.

8. When completing the data availability statement of the submission form, you indicated that you will make your data available on acceptance. We strongly recommend all authors decide on a data sharing plan before acceptance, as the process can be lengthy and hold up publication timelines. Please note that, though access restrictions are acceptable now, your entire data will need to be made freely accessible if your manuscript is accepted for publication. This policy applies to all data except where public deposition would breach compliance with the protocol approved by your research ethics board. If you are unable to adhere to our open data policy, please kindly revise your statement to explain your reasoning and we will seek the editor's input on an exemption. Please be assured that, once you have provided your new statement, the assessment of your exemption will not hold up the peer review process.

9. We note that Figure 1 in your submission contain copyrighted images. All PLOS content is published under the Creative Commons Attribution License (CC BY 4.0), which means that the manuscript, images, and Supporting Information files will be freely available online, and any third party is permitted to access, download, copy, distribute, and use these materials in any way, even commercially, with proper attribution. For more information, see our copyright guidelines: http://journals.plos.org/plosone/s/licenses-and-copyright.

We require you to either present written permission from the copyright holder to publish these figures specifically under the CC BY 4.0 license, or remove the figures from your submission:

CHANGES MADE: Permission has been granted and this statement has been included in the Figure 1 caption (Lines 89 – 91). A copy of proof of permission has been uploaded to the submission portal.

10. Please upload a new copy of Figures 4, 5, S8, S9, S10, S11, S12 and S13 as the detail is not clear. Please follow the link for more information: https://journals.plos.org/plosone/s/figures

CHANGES MADE: Figures have been modified and re-uploaded to be clearer. All figures and supplementals follow the required formats of PLOS One. Changes are described below in the reviewer comments section.

REVIEWER COMMENTS

5. Review Comments to the Author

Reviewer #1: This manuscript employs computational fluid dynamics modeling to test feeding strategies and structural stability in the enigmatic archaeocyath Yukonensis yukonensis. While the application of numerical modeling to this unique morphology is valuable, I have significant concerns regarding 1) the structure of the article, and 2) critical discrepancies between the actual fossil morphology and the geometry used for the simulation. I therefore suggest a major revision for the manuscript before publication. Of note, although I am familiar with archaeocyath, I am not a specialist in this specific taxon nor in functional morphology modeling, so I invite the authors to correct me if my interpretation of the model setup is mistaken.

1. It is difficult to follow the morphology of Yukonensis based on the current figures and text structure. Fig 1B, C does not clearly show the spine structure, yet the text describes model parameters before adequately visualizing the biological structure. As a reader, I found it confusing to reconcile the "wide" vs. "narrow" corolla descriptions (Fig 1A vs 1D) with the single species description. As a paleontologist, I believe it is necessary to fully describe the reconstructed biological structure of Yukonensis before introducing the modeling parameters, a step that is currently missing in the manuscript.

CHANGES MADE: We have added additional descriptions of the morphological terminology and forms of typical archaeocyaths earlier in the introduction to make the morphological descriptions of Y. yukonensis clearer by introducing them before discussing the models that we have constructed.

Lines 50 - 57: “In broad terms, archaeocyaths are characterized by relatively thick, porous, calcifying skeletons that range from cylindrical to conical and typically display a characteristic two-walled ‘cone-in-cone’ architecture [8]. The space between the walls is known as the intervallum and can contain a diverse array of calcium carbonate features that vertically and horizontally partition the skeleton of the archaeocyath [8]. Moreover, some archaeocyaths in the Order Monocyathida instead have a single wall and completely lack an intervallum [20]. Although these forms are often treated as the archetypal archaeocyath morphology, numerous taxa diverge markedly from this body plan [7,19–21].”

CHANGES MADE: We appreciate the reviewer’s feedback on our manuscript’s Introduction and Figure 1. Figure 1 has been modified in several ways to increase clarity. First, Fig 1B now shows a remodelled Y. yukonensis that has spines on the thorny corolla. Additionally, to reduce confusion about the wide and narrow variants, a model of the narrow variant has been added to Fig 1C. Similarly, the cross-section schematic of Y. yukonensis (Fig 1D, previously Fig 1C) also has spines included. In Fig 1C, an error where VII and VIII pointed to the same feature has also been fixed. Fig 1E is now the 2D schematic showing how the model was created along with measurements of the features. The final two panel figures (now Fig 1F and G) remain unchanged. This modified figure in combination with the glossary table (Table 1) provides further clarity on the organism and its morphology.

Lines (77-91): “Fig 1. Fossil specimens and reconstructions of Yukonensis yukonensis from the Mackenzie Mountains, Yukon, Canada. A) Longitudinal section (from Handfield [22], Plate 1, Fig 4). B) Two chambers of a reconstructed 3D Y. yukonensis model with narrow thorny corolla. C) Two chambers of a wide-angled thorny corolla Y. yukonensis reconstruction. D) Cross-sectional diagram of a single chamber. The red colour indicates internal features that were not modelled (See Materials and Methods). The labels are as follows: I) Osculum, II) Rods, III) Synapticulae, IV) Outer wall pore, V i) Spine, V ii) Connecting membrane, V) Thorny corolla, VI) T-shaped ridge (enlarged), VII) Inner wall pore , VIII) Central cavity, IX) Tumuli, X) Inner wall, XI) Outer wall, XII) Vesicle, XIII) Intervallum. E) Two-dimensional view of the thorny corolla with labelled measurements of key features. The narrow thorny corolla is labelled with ‘n’, and the wide thorny corolla variant is labelled with ‘w’, F-G) Cross-section images from Handfield and McKinney [21] displaying the intervallum, the open central cavity between chambers (E; Plate 1, Fig 1) and the aporous horizontal vesicle membrane (red arrow - F; Plate 2, Fig 2). Sections A, E, and F reprinted from [21,22] under a CC BY license, with permission from Sedimentary Geology (SEPM), original copyright [1967, 1975].”

2. There are critical discrepancies between the organism described in the text/images and the 3D geometry used for the simulations, which likely compromise the validity of the flow model results.

- The characteristic double-walled structure of archaeocyaths is clearly visible in the photomicrographs (Fig 1E, F) and the diagram (Fig 1C). However, the model (Fig 1B) lacks an inner wall entirely. The authors state this was done to "focus solely on the external structure" (lines 183-184), but for a sponge-grade organism, flow is dictated by the pressure gradient across the intervallum (Savarese, 1992). Removing the inner wall would fundamentally alter the internal flow resistance. This is particularly important given that the authors are interpreting the organism as an active filter feeder.

CHANGES MADE: Due to computational limitations, we were unable to change the model by adding the radial elements and inner wall. However, we have addressed the reviewer’s concerns regarding the conclusions of our paper on the active feeding mechanisms of Yukonensis. We have focussed our discussion on the vertical sectioning of flow and the potential functional roles of the thorny corolla, spines, and T-ridges. We have listed all our changes regarding feeding to the last comment of this review.

- The authors model the complex thorny corolla as "two-dimensional planes with no thickness" (Lines 182-183). However, the text describes the corolla as an "inverted umbrella" with distinct spines, and the authors note spines are up to 0.5 mm in diameter (Line 158, Fig. 1E). Modeling such large structures with zero thickness treats the corolla as a smooth baffle. In reality, thick spines would create surface rugosity and t

---

## [Editor Report · Decision Letter 1]

1 Apr 2026

Functional morphology of the Cambrian archaeocyath sponge Yukonensis

PONE-D-25-63578R1

Dear Mr. Zaid Ahmed Qureshi,

We’re pleased to inform you that your manuscript has been judged scientifically suitable for publication and will be formally accepted for publication once it meets all outstanding technical requirements.

Kind regards,

Shamim Ahmad, PhD

Academic Editor

PLOS One

Additional Editor Comments (optional):

Dear Author,

The revised manuscript reflects a strong and commendable effort to address reviewer concerns, resulting in a significantly improved and more coherent study. The restructuring of the manuscript, particularly the clearer introduction of morphological features prior to model implementation, has greatly enhanced readability. The revised figures are notably improved and now more effectively communicate the anatomical complexity of Yukonensis yukonensis, especially with respect to the corolla and spine structures. The methodological refinements are also appreciated. The transition from simplified two-dimensional representations to more realistic three-dimensional geometries, along with the inclusion of curved and hollow spines, represents a meaningful advancement in the modeling approach. Additionally, the decision to temper the conclusions regarding feeding mechanisms demonstrates appropriate scientific caution and strengthens the credibility of the study.

That said, one important limitation persists. The absence of the inner wall in the computational model remains a significant simplification given the fundamental role of the intervallum in archaeocyathan fluid dynamics. While your explanation and acknowledgment of this constraint are clear and appropriate, the lack of this feature still limits the biological fidelity of the simulations. If incorporating the inner wall is not feasible, it would be beneficial to further emphasize this limitation in the discussion and explicitly frame the results as representative of a simplified external-flow scenario rather than a complete organismal system. Minor issues, including previously noted inconsistencies in material properties and morphometric parameters, appear to have been satisfactorily resolved.

In summary, the manuscript is now substantially improved and scientifically more robust. With careful framing of the remaining limitation, it is well-positioned for publication.

with best wishes,

Dr Shamim Ahmad
---

## [Editor Report · Acceptance letter]

PONE-D-25-63578R1

PLOS One

Dear Dr. Qureshi,

I'm pleased to inform you that your manuscript has been deemed suitable for publication in PLOS One. Congratulations! Your manuscript is now being handed over to our production team.

Kind regards,

on behalf of

Dr. Shamim Ahmad

Academic Editor

PLOS One